# Human Chorionic Gonadotropin and Early Embryogenesis: Review

**DOI:** 10.3390/ijms23031380

**Published:** 2022-01-26

**Authors:** Sophie Perrier d’Hauterive, Romann Close, Virginie Gridelet, Marie Mawet, Michelle Nisolle, Vincent Geenen

**Affiliations:** 1Center for Assisted Medical Procreation-ULiège, Site du CHR de la Citadelle, University of Liège, 4000 Liège, Belgium; virginie.gridelet@uliege.be (V.G.); Michelle.Nisolle@uliege.be (M.N.); 2GIGA-Stem Cells, Molecular Regulation of Neurogenesis in Health and Disease, University of Liège, 4000 Liège, Belgium; 3Service de Gynécologie-Obstétrique, Site du CHU, University of Liège, 4000 Liège, Belgium; romann.close@student.uliege.be (R.C.); Marie.Mawet@doct.uliege.be (M.M.); 4GIGA-I3 Center of Immunoendocrinology GIGA Research Institute, University of Liège, 4000 Liège, Belgium; vgeenen@uliege.be

**Keywords:** hCG, implantation, pregnancy, immunology, miscarriages, thyroid function

## Abstract

Human chorionic gonadotropin (hCG) has four major isoforms: classical hCG, hyperglycosylated hCG, free β subunit, and sulphated hCG. Classical hCG is the first molecule synthesized by the embryo. Its RNA is transcribed as early as the eight-cell stage and the blastocyst produces the protein before its implantation. This review synthetizes everything currently known on this multi-effect hormone: hCG levels, angiogenetic activity, immunological actions, and effects on miscarriages and thyroid function.

## 1. Introduction

The relationship between the pituitary gland and the reproductive organs was established in the early 20th century. In 1927, Ascheim and Zondek (a gynecologist and endocrinologist, respectively) demonstrated that the blood and urine of pregnant women contained a gonad-stimulating substance, and human chorionic gonadotropin (hCG) was discovered [1].

hCG is an essential hormone secreted by the trophoblast in its early development. hCG has a number of actions that aim to preserve the embryo, including progesterone secretion maintenance by the corpus luteum [2]. 

hCG is a glycoprotein hormone ranging from 36 up to even 41 kDa from low to highly glycosylated forms. It is composed of two subunits, α and β linked with a non-covalent bond [3]. Their properties are summarized in Figure 1. 

hCG has a half-life of 24 to 36 h and is mostly catabolized by the liver. One-fifth is also excreted in the urine after degradation to subunits and nicked forms, especially hCGβ core fragment (hCGβcf), a metabolite produced via the degradation of hCG [4]. 

The measurement of hCG and its related molecules is useful in clinical practice, but larger awareness is needed worldwide regarding the use of new sensitive and specific assays tailored to different clinical applications [5].

Extragonadal gonadotropin action has always been a controversial topic. Rivero-Müller and Huhtaniemi have said that the physiological function of the LH-hCG receptor (LHCGR) outside the gonads seems doubtful and superfluous, at least as concerns female reproduction [6].

The hormone hCG is specific to humans and the hormone eCG is specific to the equine species. In all other mammals, this hormone is replaced by the hormone LH. In veterinary clinical practice, the injection of hCG and eCG hormones is widely administered in the reproduction procedures of domestic mammal species (sows, cows, even female mice, etc.). In this case, hCG is used as an analog of LH and eCG because it mainly shows FSH activity [7].

The objective of this review is therefore to present hCG isoforms and their functions—hCG being the basis of reproduction, angiogenesis, and immunology in early pregnancy. We will explore some clinical advances and implications of using hCG, particularly in the IVF or embryo transfer procedures, and in miscarriages. hCG has many similarities with TSH in early pregnancy that we will present at the end of the article. 

## 2. hCG Isoforms and Secretion

Ulf-Håkan Stenman et al. describe the biochemical and biological background for the clinical use of determinations of various forms of hCG [8].

hCG exists in four forms, known as classical hCG, hyperglycosylated hCG, the free β unit of hyperglycosylated hCG, and sulphated hCG [9]. Each of these four molecules has different physiological functions (Figure 2). 

### 2.1. The Classical hCG 

This is one of the first molecules secreted by the embryo. Its RNA is transcribed as early as the eight-cells stage [10] and the blastocyst produces the protein before implantation [11,12]. During the implantation, hCG is mainly secreted by the syncytiotrophoblast and less by the cytotrophoblast. Clinical biology can thus detect hCG in the maternal blood 10 days after ovulation. Its concentration reaches its top level around the 10th and 11th weeks of gestation. Afterwards, this level decreases and remains basal from the 12th week of gestation onwards until the end of the pregnancy. However, it remains significantly higher than in non-pregnant women [13,14].

By binding to his receptor called LHCGR, classical hCG acts on multiple types of cells: corpus luteum cells, myometrial smooth muscle cells, endothelial cells, and decidual cells. 

During the fifth and sixth days of embryogenesis, the blastocyst secretes hCG into the uterine cavity. This hormone binds to its LHCGR on the deciduous surface. In response, the decidua prepares for implantation [15,16,17]. hCG influences stromal cells by underpinning the decidualization and the prolactin secretion [18]. We have shown in our laboratory that the hCG–LHCGR complex also increases the secretion of leukemia inhibitory factor (LIF) and decreases the secretion of interleukine-6 (IL-6) by endometrial cells, factors affecting embryo implantation [16]. This complex also promotes the differentiation of cytotrophoblasts into syncytiotrophoblasts [19]. The hCG–LHCGR complex also regulates prostaglandin synthesis [20] and the formation of cAMP [21]. hCG encourages trophoblast invasion and interstitial theca cell proliferation by over-modulating ERK and AKT signals [22,23]. 

Aside from this hCG-LHCGR complex, it has been shown that multiple hCG isoforms could stimulate trophoblastic invasion without regard to the LHCGR [24]. 

As said above, hCG plays an important role in synchronizing fetal and endometrial developments. Throughout pregnancy, hCG is also a marker of placental function. 

### 2.2. Hyperglycosylated hCG 

hCG-H β subunit has four oligosaccharide-linked Os instead of two in the classical form of the hCG β subunit [25]. It is massively produced during the first trimester of pregnancy, particularly by the extravillous cytotrophoblasts.

It represents the majority of the total hCG in the third week of gestation and the half during the fourth week. Then, it decreases rapidly until it completely disappears from the maternal blood circulation at the end of the first trimester [26]. hCG-H is useful for predicting pregnancy outcomes in women, with a first trimester suspicion of abortion. Nowadays, it is not considered as a better tool than the classical form of hCG [27].

hCG-H acts through autocrine instead of endocrine action. It decreases the apoptosis of trophoblast cells [28] and induces the implantation of the embryo [29] and trophoblastic invasion [30]. hCG-H is also massively secreted by choriocarcinomas and germ cell tumors [25,30,31]. Its anti-apoptotic action would be achieved by its binding with the TGF-β receptor and independently of LHCGR. hCG-H monitoring is useful in predicting Down’s syndrome [30], pre-eclampsia [32], and the therapeutic response to trophoblastic diseases, as well as in pregnancy predictions performed in in vitro fertilization [33].

### 2.3. The Free β Subunit 

This subunit acts as an agonist of LHCGR and an antagonist of the TGF-β receptor. Gestational hypertension could also be predicted by the abnormal rise in the circulating free β subunit of hCG. However, the association of β-hCG and inflammation, and oxidative stress in a pregnancy-caused hypertensive disorder, on the perinatal stage remains unclear. However, Wang et al. demonstrated via a case–control study that the correlation of circulating free β subunit levels with inflammatory and oxidative stress markers in patients with pregnancy-induced hypertension in perinatal stage was statistically significant [34]. 

Like hCG-H, maternal serum free β-hCG is also used as a biomarker in first trimester screening for fetal Down’s syndrome [35]. The free β subunit also has a promotive action on cancer: germ cell malignancies, epithelial malignancies or carcinomas, adenocarcinomas, sarcomas, teratomas, blastomas, leukemias and lymphomas [36]. For example, this action on the bladder carcinoma is exerted through the inhibition of apoptosis [37]. According to P. Sirikunalai et al., abnormally low (<0.5 MoM) or high (>2.0 MoM) free β subunit levels during gestation are generally associated with an increased risk of adverse pregnancy outcome (spontaneous abortion, preterm birth, low Apgar score, etc.) [38]. 

### 2.4. The Sulphated hCG 

This isoform is produced by the pituitary gland in non-pregnant women and is secreted at the same time as LH during the cycle. Hence, its concentration ranges around one-fiftieth of the LH concentration [39,40,41,42]. While these levels are low, sulphated hCG is exactly 50 times more potent than LH [43] and acts the same way by stimulating androstenedione production during the follicular phase of the cycle as well as stimulating ovulation and corpus luteum formation. During the luteal phase, it may help stimulate progesterone production [39,40,41,42].

## 3. hCG Levels and Pregnancy

Ascheim and Zondek discovered hCG while injecting this substance into immature female mice. This injection produced follicular maturation, luteinization and hemorrhage into the ovarian stroma. The first specific test was created and was known as the Ascheim Zondek pregnancy test [1].

Nowadays, hCG levels can be detected by serum or urinary testing. Most over-the-counter urine assays are now based on antibodies targeting hCG, hCGβ and hCGβcf, which are the largest molecular variants of hCG in urine. hCG returns to zero from 7 to 60 days after delivery or abortion [3,13]. 

In everyday clinical practice, hCG is mainly used to diagnose pregnancy and to supervise first trimester adverse pregnancy outcomes. Abnormalities in the production and the circulating levels of hCG during specific periods of gestation have been associated with a large array of pregnancy complications, such as miscarriages [38], fetal chromosomal anomalies [43], pre-eclampsia [44,45], disturbances in fetal growth and development [46] and gestational trophoblastic diseases [47].

Nevertheless, the persistence of low hCG concentrations in a non-pregnant woman is not always malignant and can even be benign. Therefore, the clinician must identify the reason first, before prescribing any treatment [40,48]. 

In addition, very high concentrations of hCG have been shown to have deleterious effects on fetal tissues, notably on fetal gonadal steroidogenesis [49]. To avoid this, the human fetal tissue macrophages are thought to eliminate excess hCG. Katabuchi et al. have shown that hCG induces the formation of vacuoles in human monocytes and hypothesized that these vacuoles would be involved in the protection of fetal tissues [50].

Multiple factors influence hCG levels during pregnancy. Among them, endocrine disruptive chemicals (EDCs), particularly bisphenol A and para-nonylphenol, can modulate hCG production and cause fetal damage as well as long-lasting consequences in adult life [51].

## 4. Angiogenic Actions of hCG

Classical hCG has angiogenic actions through the LHCGR and achieves many of its functions through the regulation of the expression of endocrine gland-vascular endothelial growth factor (EG-VEGF) and its receptors [52,53,54].

hCG increases blood vessel formation and the migration and maturation of pericytes in different in vitro and in vivo models. Through this action, the trophoblast can form plugs that prevent maternal blood from bleeding into the intervillous spaces during early pregnancy [54,55,56,57,58].

It also enhances the secretion of VEGF through the activation of NF-κB on angiogenesis during the luteal phase [55,59,60]. In addition, hCG shields vascular endothelial cells against oxidative stress through the inhibition of apoptosis, activation of cell survival signaling, and mitochondrial function retention [61]. Jing et al. have shown that the decreased production of the β subunit in early pregnant women could act on the expression of VEGF-MEK/ERK signal pathway by down-regulating it. It reduces angiogenesis and eventually leads to the abnormal angiogenesis of the villosities, a mechanism which may be an important factor of missed abortion [62].

As hCG, hCG-H still presents a potent angiogenic effect but is acting regardless of LHCGR signaling pathways [63,64]. Gallardo et al. have suggested that the striking overlapping of hCG and Heme oxygenase-1 (HO-1) functions in pregnancy could indicate that hCG hormonal effects are mediated by HO-1 activity, which may be affected by a HMOX1polymorphism in humans [65].

hCG and its hyperglycosylated isoform are accordingly considered pro-angiogenic molecules granting adequate fetal perfusion and fetal-maternal exchanges.

## 5. Immunological Actions of hCG

The immunomodulatory properties of hCG are various and important for maternal tolerance of the embryo, an essential mechanism for the embryonic implantation and development [53,66,67].

Obviously, immune cells situated in the uterine cavity play a key role in the embryo implantation [68,69].

### 5.1. Th1/Th2/Th17/Treg Paradigm

CD4 + T cells can be classified into the following subsets: T helper (Th) Th1, Th2, Th17 and regulatory T cells (Treg), according to their functions.

Raghupathy R et al. indicated that the Th1/Th2 hypothesis was relevant to women suffering from recurrent miscarriages [70]. Even so, the Th1/Th2 paradigm has now been enhanced to the Th1/Th2/Th17 and Treg paradigm widely accepted.

Th17 and Treg cells are implicated in the process of autoimmune diseases and infection. Therefore, previous studies have shown that Th17/Treg imbalance can also be associated with recurrent spontaneous abortion [71,72].

hCG modulates the balance between inflammatory type Th1/Th17 cells and anti-inflammatory type Th2/Treg cells, and therefore plays a fundamental role in the implantation of the embryo [53,73].

### 5.2. T-Cell

hCG inhibits T-cell proliferation [74], but trophic actions of hCG have also been reported [75]. It has been shown that hCG could interact with the T-cell receptor (TCR) signal [76].

hCG might stimulate CD4 + 25 + T cells proliferation by attracting these cells to the endometrium in early pregnancy [77,78].

On murine Treg cells, hCG increases their frequency in vivo, and decreases their suppressive activity in vitro, which can scale down the rate of miscarriages for example [67,79]. On the human model, peripheral Treg cells percentage was increased when intrauterine hCG was administered [80,81].

hCG increases the presence of regulatory T cells and the level of IL-1β in mice [80,81]. It also increases the production of IL-2 by naïve and memory T cells which regulates these very T cells. hCG decreases the expression of CD25 and CD28 on the surface of naïve T cells (CD45RA+) and the expression of CD25 on memory T cells (CD45R0+). It seems that hCG encourages the differentiation of memory T cells by improving the expression of CD45R0+ and decreases their functional activity towards fetal antigens through a competitive mechanism [82].

### 5.3. Uterine Natural Killer Cells

hCG has a beneficial effect on uterine natural killer cells (uNKs), significant leukocyte cells in a non-gestating uterus that act on the formation and survival of embryo implantation in women and mice [83,84,85,86].

hCG regulates the proliferation of uNKs [87] in a dose-dependent manner in vitro. These cells do not express LHCGR and hCG would act directly on these cells through the mannose receptor [88]. UNKs plays a role in the remodeling of spiral arteries, which guarantees a satisfactory supply to the fetus by the placenta [89]. As outlined above, they also secrete proangiogenic molecules such as members of the VEGF family [90].

### 5.4. Bone Marrow-Derived Dendritic Cells

In a murine model, different teams have demonstrated an inhibitory effect of hCG on dendritic cells (DCs) as well as on peripheral and local (decidual) DCs in a way that hCG supports a tolerogenic rather than an immunogenic DC phenotype. Furthermore, hCG influences the differentiation and function of DCs, decreasing their ability to stimulate T-cell proliferation [73,91,92].

### 5.5. Monocytes and Macrophages

hCG acts on the monocytes by promoting their function and secretion of IL-8 [93] and induces the functions of macrophages [94]. Thus, hCG induces the cleaning of the endometrium by purifying apoptotic cells and fighting possible infections, two important mechanisms in the maintenance of pregnancy.

### 5.6. Other Immunological Molecules

It has been shown that hCG could increase the ability of trophoblast cells to invade the extracellular matrix in vitro. It is accompanied by an increase in the expression of the matrix metalloproteinases (MMP)-2 and -9 and VEGF. It is also accompanied by a decrease in the expression of the tissue inhibitor of metalloproteinases (TIMP)-1 and -2. Peripheral blood mononuclear cells (PBMC) support in vitro embryo invasion and hCG enhances the effects of PBMCs [95].

In vitro, hCG is not a regulator of cell damage from PMBCs. Though, in an inflammatory context, hCG appears keeping the delicate balance between plasmacytoid dendritic cells and myeloid dendritic cells (MDC) and seems to retain a tolerogenic MDC1profile [96].

When administrated, hCG predominantly manages the cytokine profile of the endometrium [97]. Cytokines are known to play a key role in the female immune response during conception, implantation, maintenance of pregnancy, embryo development, etc. [98]. hCG directly or indirectly influences the genetic expression of several cytokines in cell signaling, proliferation, apoptosis, immunological modulation, tissue remodeling as well as angiogenesis in endometrial stromal cells [99]. A study performed in a 3D cell culture model established that hCG administration substantially alters the production of several cytokines in epithelial cells, stromal cells and both cell types together [100].

hCG reduces the expression of TNF-α and INF-γ in the maternal-fetal interface and decreases the rate of resorption in abortive mouse models [101]. Bai et al., cultured in vitro PBMCs and showed that hCG significantly inhibited IL-6 and TNFα mRNA expression, indicating that hCG could inhibit the production of proinflammatory cytokines [102].

Control of complement system’s activation in the fetal-maternal environment is critical for embryo development. hCG influences the decay accelerating factor (DAF) and the C3 protein in in vitro and in vivo models [103].

Consequently, hCG has an important immunomodulatory function by its effects on uNK cells, Treg, Th1/Th2/Th17, DC, and macrophages (Figure 3).

## 6. Infusion of hCG during Embryo Transfer

Given its multiple roles in the early stages of pregnancy, hCG was thought to possibly improve the outcomes of patients with a history of repeated implantation failure (RIF). RIF is defined as an assisted reproductive technologies (ART) failure after at least four unsuccessful transfers of good-quality embryos, or at least four cleaving stages or at least two failed blastocyst transfers. The causes of RIF are attributed to either poor-quality embryos or to defective endometrial receptivity [104]. Unfortunately, in many patients, the etiology of RIF is not identified.

The implantation of an embryo requires a certain degree of inflammation at the endometrial implantation site. This inflammation is mainly mediated by T-cells and macrophages. It is thought that, in patients with RIF, a lack of inflammation at the implantation site could explain the absence of successful implantation. This hypothesis is confirmed by studies conducted in RIF animal models in which PBMCs were instilled in the uterus before embryo transfer and seemed to improve pregnancy rates. In the recent meta-analysis of Qi Qin et al. [105], the pregnancy and live birth rates were significantly increased in the group of RIF women treated with an instillation of PBMCs cocultured with hCG compared with the control group. In summary, pregnancy rates seemed to improve when hCG was cultured with PBMCs.

A problem from this technique is that exaggerated inflammation was also shown to be detrimental to the continuation of the pregnancy. It was suggested that, when co-cultured with PBMCs, hCG could balance the immunologic response to PBMCs. Several clinical studies, with different study designs, have tested this hypothesis with relatively good results [106].

Moreover, Tesarik et al. showed that hCG administration to recipients amplify endometrial thickness on the day of embryo transfer and improved endometrial receptivity [107,108].

To increase the success rates of embryo transfer during IVF, isolated hCG intrauterine infusion has been suggested but the results of the studies are questionable [105,106,109,110,111,112,113,114].

Bielfeld et al. demonstrated that the endometrial proteome composition of RIF patients differs from fertile controls during the window of implantation. The in vivo infusion of hCG into the uterine cavity of RIF patients stimulated the presence of endocytosis proteins, hypoxia-inducible factor 1 (HIF1) signal and chemokine production [115].

Infusion of hCG during the implant window in a non-human primate model increased the expression of glycodelin, an immunomodulator protein secreted by the glandular portion of the endometrium [116,117].

A recent randomized clinical trial study made by Hosseinisadat et al. showed that hCG intrauterine injection after oocyte retrieval does not improve implantation, chemical or clinical pregnancy rates in ART cycles. Further studies are necessary to clearly identify the role of hCG intrauterine injection in the day of oocyte retrieval in ART outcomes [118].

## 7. hCG and Miscarriages

The hCG molecule is an extremely important multifaceted hormone involved in hormonal interactions of the fetal–placental–maternal unit, as well as neuroendocrine and metabolic changes that occur in the mother and in the fetus during pregnancy and at parturition, as well as pathophysiologic functions in non-pregnant women: a potential biomarker for preeclampsia, a serum marker for down’s syndrome screening, and a crucial marker in the diagnosis of gestational trophoblastic disease. More investigations on the precise role of hCG and its pathophysiologic functions need to be explored [108].

A study showed that analyzing hCG and progesterone can predict the evolution of the pregnancy in 41.1% of patients [119]. It is also known that hCG, hCG-H and progesterone levels may be predictive for pregnancy outcomes in patients with recurrent miscarriages 14 days after oocyte retrieval and 11 days after embryo transfer [120,121,122].

Recurrent spontaneous abortion (RSA) is one of the most common complications of early pregnancy. This affects about 10 to 20% of all pregnancies occurring predominantly during the first 12 weeks of pregnancy [123]. The causes of embryo loss are multifactorial and include cytogenetic abnormalities, maternal problems (e.g., lupus erythematosus or diabetes), uterine malformations, cigarette smoking, inadequate placental development, etc. [124,125]. One study showed that hCG-related cytokines, MIP1a/hCG, GCSF/IL-1ra, and MIP1a/TGF-β1 ratios after 4 weeks of pregnancy were significantly altered in women with spontaneous miscarriages [79]. In the decidual and placenta of pregnant women with RSA, Schumacher et al. found that levels of hCG and Treg cells are reduced compared with a normal pregnant woman [67].

A meta-analysis of five studies aimed to demonstrate whether hCG treatments could prevent miscarriage. The results of these studies remain uncertain [124].

The combination of hCG and immunoglobulin treatment on Th17 + cells and Foxp3 + Treg cells in patients with RSA was examined and the Th17/Treg ratio was decreased, which could be beneficial for these patients [126]. Another study inspected the impact of hCG in the regulation of Foxp3 + p cells in patients with RSA and the results suggest that this regulation may have a positive impact on the pregnancy outcomes [110].

## 8. hCG and Thyroid Fonction

As displayed in Figure 1, hCG and TSH have the same alpha subunit. Although different at first glance, they converge during the first trimester of pregnancy.

The reason of the potent stimulatory effects of hCG on gestational thyroid function remains unknown and continues to be explored. Low maternal thyroid function and impaired thyroidal response to hCG stimulation during the first trimester are associated with a lower ultrasound crown–rump length (CRL). These data can help to improve the identification of pregnancies at high risk of fetal growth restriction and adverse pregnancy or child outcomes [127,128].

Early stage pregnancy serum β-hCG levels measured on 14th day after oocyte pickup were lower and the miscarriage rate was higher in patients with thyroid autoimmunity. Thyroid hormonal dysfunction and thyroid autoimmunity are associated with a higher risk of adverse pregnancy outcomes during the entire pregnancy [128].

Korevaar et al. showed that the risk of premature delivery according to TSH concentrations was modified by hCG concentrations. Thyroid peroxidase antibody (TPOAb) positivity can attenuate gestational thyroid responses to hCG during pregnancy. The effects of thyroglobulin antibodies (TgAb) remain unknown, but their results imply that TgAb, in addition to TPOAb, could also interfere with thyroidal responses to hCG during the first half of pregnancy. These data suggests that the assessment of maternal thyroid function together with hCG concentrations may improve the risk of premature delivery and give insights into the pathophysiology of the association between maternal thyroid function and premature delivery [129].

hCG stimulates the maternal thyroid gland, and maternal thyroid function can be associated with the pathophysiology of gestational diabetes mellitus (GDM) [130].

## 9. Conclusions

Human chorionic gonadotropin (hCG) is a multi-effect hormone which has an incredible impact on humans and more specially on the acceptation and the success of the gestation. The cytokine–hCG interaction is well known and cytokines plays a key role in the female immune response during conception, implantation, maintenance of pregnancy, embryo development, etc.

hCG acts through different pathways and on multiple cell types. It promotes the acceptation of the embryonic implantation, angiogenesis and vasculogenesis, and the control of the trophoblast differentiation, as well as the immune regulation of the maternal-embryonic or fetal interface during the entire pregnancy. hCG plays a key role in the IVF or embryo transfer, in miscarriages, and it can be correlated with thyroid function.

## Figures and Tables

**Figure 1 ijms-23-01380-f001:**
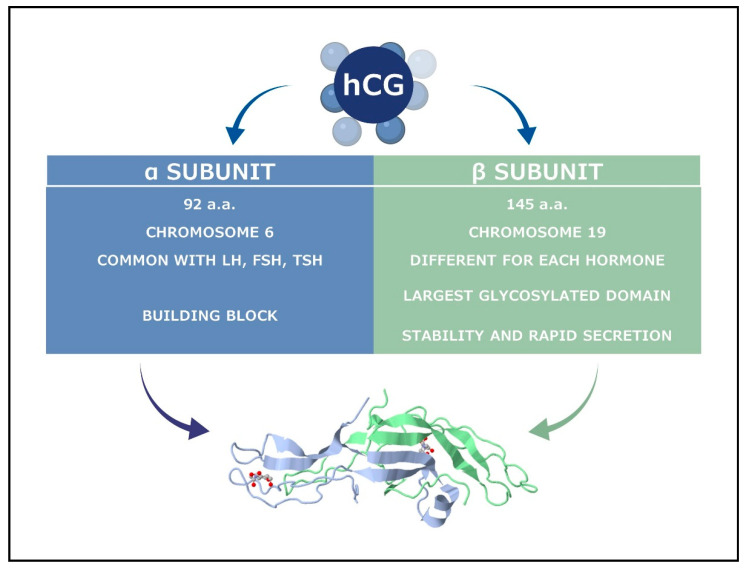
Structure of hCG at 2.6 angstrom resolution from MAD analysis of the selenomethionyl protein (JSMol Viewer: modern web app for 3D visualization and analysis of large biomolecular structures, RCSB PDB, doi: 10.1016/s0969-2126(00)00054-x. PMID: 7922031). In blue, the α subunit and in green, the β subunit.

**Figure 2 ijms-23-01380-f002:**
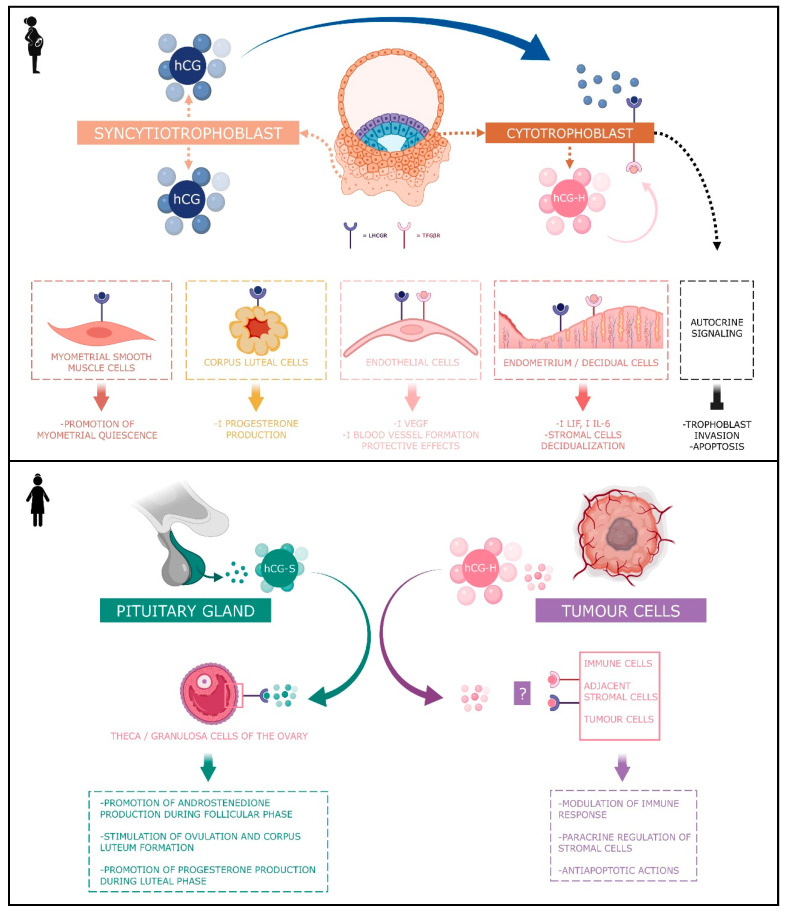
Schematic view of the actions of different forms of hCG during pregnancy and in non-pregnant woman. The classical form of hCG is schematized by a blue dot, the hyperglycosylated by a pink dot and the sulphated form of hCG by a green dot. The blue receptor is the LHCGR and the pink receptor is the TGFβR.

**Figure 3 ijms-23-01380-f003:**
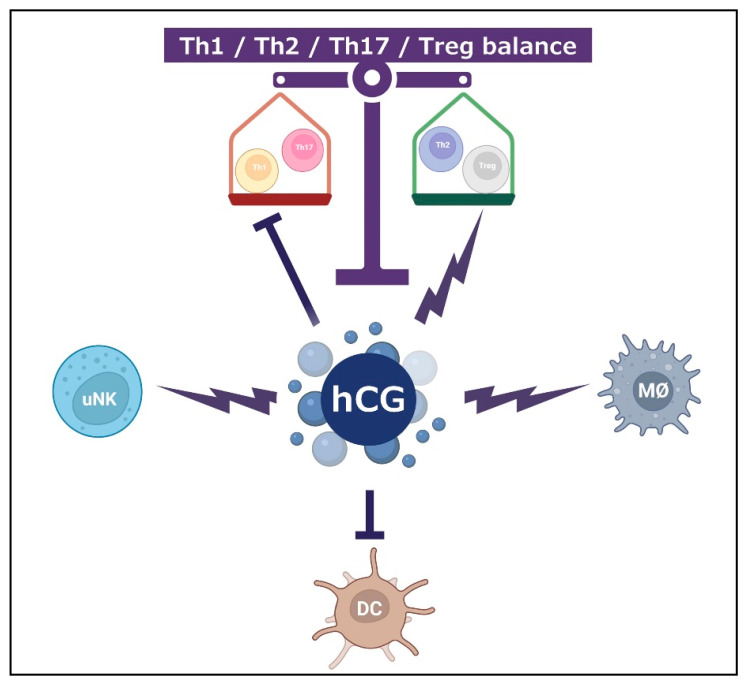
hCG and its immunotolerance effects through multiple cell types. hCG is acting positively on uNK, macrophages (MØ), and Th2/Treg. hCG is acting negatively on DC, and Th1/Th17.

## Data Availability

Not applicable.

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
