# Peer review of "Human Chorionic Gonadotropin and Early Embryogenesis: Review"

_ijms, 2022, doi:10.3390/ijms23031380_

Round 1
Reviewer 1 Report
Comments and Suggestions for Authors
The study entitled “Human chorionic gonadotropin and early embryogenesis: Review” is well written and easy to understand. This is an interesting review, which includes sections that the hCG hormone play an important role. Three main points should be revised: (i) Authors should take account the general considerations of IJMS-Manuscript preparation regarding to acronyms/abbreviations, (ii) All references should be as an only format, (iii) Conclusions should be rewritten.
Remarks to the author:
Introduction
- The hCG is a glycoprotein hormone ranging from 36 even to 41 kDa (from low to highly glycosylated forms). Cite: Cole LA. Hyperglycosylated hCG, a review. Placenta. 2010 Aug;31(8):653-64. doi: 10.1016/j.placenta.2010.06.005. Epub 2010 Jul 8. PMID: 20619452.
- The summary properties showed in the figure 1 is very clarified but, could be the authors enhances the figure resolution? Moreover, could be cite the structure of hCG obtained of RCSB PDB? Cite. doi: 10.1016/s0969-2126(00)00054-x. PMID: 7922031. The authors can download a hight resolution image in the follow linke https://www.rcsb.org/structure/1hcn.
- The sentence: “Chorionic gonadotropins exist in primates (in humans, this is hCG) and in equines (eCG). In the pregnant mice, hCG is substituted by LH (3)” should be modified by a clear sentence indicating that the hormones hCG and eCG are only secreted in the human and equine species, respectively. In addition, for clarification to authors, it should be noted that in veterinary clinical practice the injection of hormones hCG and eCG are widely used in reproductive procedures of domestic mammalian species (sow, cow, even female mice...), the hCG hormone used as LH analogue and eCG because it mainly shows FSH activity. The authors should not confuse this with the fact that hCG is replaced by LH secretion in pregnant mice, it is not that. In the study by Gridelet et al. 2013 the female mice were treated with an injection of hCG to simulate an effect of LH.
The classical hCG
- “Clinical biology can thus detect hCG”
- Unify the ordinal numbers with or without superindex.
Hyperglycosylated hCG:
- “HCG-H β subunit has four oligosaccharide-linked Os instead of two in the classical form of hCG”. HCG-H β subunit or hCG-H?
HCG levels and pregnancy:
- hCGβ Core fragment. Please, define this acronym.
- “Many over-the-counter urine pregnancy tests do not detect hCG-H”.
Conclusions:
- "Human chorionic gonadotropin (hCG) is a multi-effect hormone and a cytokine ..." The conclusion should be rewritten.
The hCG hormone is not a cytokine.
The cytokine-hCG interaction is well known, but this fact does not make it a cytokine. Mounting evidences affirms that cytokines are molecules that play a key role in the female immune response during conception, implantation, pregnancy maintenance, embryo development ... (Robertson SA, 2015) On the one hand, the expression of cytokines may be conditioned by the hormone hCG (Carp HJ, 2010), and on the other hand, cytokines (mainly derived from Th2) induce the release of hCG (Saito S, 2000).

Reviewer 2 Report
Title: “Human chorionic gonadotropin and early embryogenesis: Review”
By Perrier d’Hauterive et al.
In opinion of this reviewer, this is a very interesting review for International Journal of Molecular Sciences. Information presented here on hCG and embryogenesis provides an updated basis for both basic and clinical knowledge. The manuscript is well presented. However, some points should be addressed:
1) A short historical outline of the development of research on the function and clinical use of hCG or their subunits should be included in the introduction section. In this sense, I suggest to the authors to read the reviews Hum Reprod Update 2004, 10: 453–467; Hum Reprod Update 2006, 12: 769–784; and Hum Reprod Update 2009, 15: 69–95. Furthermore, hCG is extensively used in the control of reproduction in domestic animals. These points would improve the manuscript.
2) The objective of this review should be included at the end of the introduction.
3) Authors should explore at least some clinical advances/implications of using hCG, particularly in the IVF or embryo transfer procedures, and in miscarriages.
Round 2
Reviewer 1 Report
All my previous concerns have been addressed in this revised version of the manuscript.As minor correction I would still suggest:
Check the meaning of the sentence in lines 59-60. "In all other mammals this hormone is not secreted, but the hormone LH only". Is secreted only LH hormone? In my point of view should be rewritten.
Author Response
Thank you very much for the review!
Your minor review was considered and the lines 59-60 was modified this way: "In all other mammals this hormone is not secreted, but is replaced by the hormone LH".
Yours sincerely
Reviewer 2 Report
The manuscript has been much improved. I would like to thank the authors for considering my comments and suggestions.
Author Response
Thank you very much for your review.
Yours sincerely.